# Interaction of Zinc Mineral Nutrition and Plant Growth-Promoting Bacteria in Tropical Agricultural Systems: A Review

**DOI:** 10.3390/plants13050571

**Published:** 2024-02-20

**Authors:** Arshad Jalal, Enes Furlani Júnior, Marcelo Carvalho Minhoto Teixeira Filho

**Affiliations:** School of Engineering, Department of Plant Health, Soils and Rural Engineering, São Paulo State University (UNESP), Ilha Solteira 15385-000, SP, Brazil; enes.furlani@unesp.br

**Keywords:** Zn fertilization, microbes-mediated biofortification, inoculation, co-inoculation, Zn-use efficiencies

## Abstract

The relationship between zinc mineral nutrition and plant growth-promoting bacteria (PGPB) is pivotal in enhancing agricultural productivity, especially in tropical regions characterized by diverse climatic conditions and soil variability. This review synthesizes and critically evaluates current knowledge regarding the synergistic interaction between zinc mineral nutrition and PGPB in tropical agricultural systems. Zinc is an essential and fundamental micronutrient for various physiological and biochemical processes in plants. Its deficiency affects plant growth and development, decreasing yields and nutritional quality. In tropical regions, where soil zinc availability is often limited or imbalanced, the PGPB, through different mechanisms such as Zn solubilization; siderophore production; and phytohormone synthesis, supports Zn uptake and assimilation, thereby facilitating the adverse effects of zinc deficiency in plants. This review outlines the impacts of Zn–PGPB interactions on plant growth, root architecture, and productivity in tropical agricultural systems. The positive relationship between PGPB and plants facilitates Zn uptake and improves nutrient use efficiency, overall crop performance, and agronomic biofortification. In addition, this review highlights the importance of considering indigenous PGPB strains for specific tropical agroecosystems, acknowledging their adaptability to local conditions and their potential in sustainable agricultural practices. It is concluded that Zn fertilizer and PGPBs have synergistic interactions and can offer promising avenues for sustainable agriculture, addressing nutritional deficiencies, improving crop resilience, and ensuring food security.

## 1. Introduction

The rising global population, loss of land resources, and climate change have severe consequences on agriculture production systems, leading to food and nutrition insecurity [1,2]. Global crop improvement programs have been given satisfactory attention to ensure food security; however, there has been little research in modern breeding and genetics, and imbalanced fertilization, which are the primary causes of nutritional insecurity in different staple crops [2,3]. Most of the global population is deficient in essential micronutrients, which increases the demand for sustainable agriculture practices for higher quality and yield of crops. Persistent micronutrient deficiency is one of the emerging health and social concerns of present and prospective populations; it is described as a hidden hunger affecting more than 2 billion, or one-third of individuals worldwide [4,5]. The above-mentioned deficiency of essential micronutrients, especially zinc (Zn), is a silent epidemic that affects about 17% of the global population and is considered to be the 5th global health risk 43 factors [6]. In addition, around 50% of cereal-cultivable soils show widespread micronutrient deficiency, including Zn deficiency, due to high soil pH; oxides; complex insoluble carbonates; and bicarbonates [7,8]. Most of the soils deficient in Zn either have low available Zn or a form that is unavailable to plants, (depending on the nature of the soil—such as calcareous, paddy, sandy, and coarse-textured soils), with elevated levels of phosphorous and silicon [9]. Therefore, Zn fertilization is considered one of the feasible strategies that improve nutrient availability in soil and promote the growth and productivity of crops, leading to agronomic biofortification [10,11].

Increasing awareness regarding the proper application of micronutrient fertilizers, coupled with numerous research initiatives in recent decades, has led to a reduction in malnutrition in the global population [12,13]. However, satisfying the requirements of a rapidly increasing population with high levels of soil fertilizers can deteriorate human health and environmental safety [14]. Therefore, adopting sustainable agricultural practices becomes crucial for ensuring global food and nutritional security with high economic benefits, by protecting and promoting the ecosystem and reducing environmental risks [15]. 

The translocation and distribution of nutrients depend on the bioavailability of nutrients in soils, plant species and the microbial population in the rhizosphere [16]. In this context, it has been discussed that plant growth-promoting bacteria (PGPBs) multiply in the rhizosphere of plants to promote soil fertility, crop productivity, and nutrient bioavailability, to deal with food security under sustainable agricultural management practices [17]. A diverse group of PGPBs, including species of *Rhizobium*, *Pseudomonas*, *Azospirillum*, *Azotobacter*, *Gluconacetobacter*, *Bacillus*, *Burkholderia*, *Klebsiella*, *Enterobacter* and many others are studied to promote plant growth and nutrient bioavailability [13,18,19]. These PGPBs affect several ecological processes, including the decomposition of organic matter, homeostasis, and nutrient cycling, that can improve crop growth and health [20]. In addition, some other processes such as nitrogen fixation; nutrient solubilization; enzyme activation; and phytohormone production (direct mechanisms), as well as siderophore production; and inhibition of phyto-pathogens (indirect mechanisms), are responsible for better growth, yield, and nutrient acquisition [21]. Plant growth-promoting bacteria play a significant role in root elongation and root physiology, enhancing the absorption of essential nutrients, particularly Zn, and contributing to improved plant growth and increased yield of different crops [22,23].

Inoculation with PGPBs under adequate fertilizer management is a promising sustainable alternative option that can improve the nutritional status and productivity of different crops. However, there is a lack of literature on the study of the integrated use of PGPBs, soil, and foliar Zn in cereal and legume cropping systems under tropical savannah conditions. Based on field-conducted research, this review focused on the concentration of targeted nutrients (Zn) in the grains of staple food crops to eliminate Zn malnutrition. Based on the mechanisms of actions of PGPBs in plants, it is expected that there would be a synergistic interaction between PGPBs and Zn fertilization, in either soil or foliar application, for improving the nutrition and productivity of cereal and legume crops. In this scenario, the integrated use of inoculation/co-inoculation of PGPBs with soil and foliar Zn fertilizer application was focused on, to highlight their impact on crop productivity and Zn daily intake to stimulate nutrient availability for obtaining the desired nutrient-enriched grain crops. In this context, the objectives of this review were to evaluate the effect of seed inoculation/co-inoculation with PGPBs in combination with Zn fertilizer application on the growth, yield, and nutritional status of cereal and legume crops. An additional goal was to investigate the effect of inoculation/co-inoculations and Zn fertilization on Zn use efficiencies for sustainable agriculture in tropical savannah regions.

## 2. Dynamics and Management of Zinc

Zinc (Zn) deficiency occurs due to low Zn distribution and solubility in soil solutions that lead to global Zn deficiency in crop plants, resulting in stunted growth and malformed leaves, reducing crop productivity, and producing malnourished grains and fruits [24]. The global Zn deficiency index indicated that the tropical regions still experience the prevalence of Zn deficiency in humans and soils (Figure 1).

Zinc is an immobile nutrient in the soil solution and its mobility is further reduced by high soil pH; calcium content; intensive agricultural practices; excessive use of fertilizers; and fertilizer adsorption to soil colloids. Zinc deficiency is the most frequent micronutrient deficiency on the surface of the soil, which restricts cereal production in a tropical agricultural system. Many tropical soils have undergone extensive weathering that may have caused the decomposition of all primary minerals (such as Zn and other micronutrients) from sedimentary and igneous rocks, leaving soil deficient in both macro and micronutrients. For example, the tropical regions of Latin America (the Cerrado region of the central plateau of Brazil, the Llanos Orientales of Columbia, and Venezuela, as well as in Costa Rica, Guatemala, Mexico, and Peru) are reported as Zn deficient regions [26]. In addition, higher phosphorous demand in tropical agriculture is another major constraint that can hinder the availability and functions of several micronutrients, especially Zn, stimulating reactive oxygen species and affecting crop growth and yield [26]. 

Zinc is an essential and important nutrient for the normal growth and reproduction of crop plants, a small amount of Zn is required for the proper physiological, enzymatic and metabolic processes of plants [27]. Zinc is a structural component and regulatory cofactor of several enzymes and proteins of different biochemical pathways, such as carbohydrate synthesis; photosynthesis; protein metabolism; pollen formation; cell membranes; and resistance to infection by certain pathogens [27,28]. Zinc influences root development, architecture, and biomass, and is transported through symplast pathways through a complex interplay of different mechanisms. Zinc transportation is facilitated via the xylem to the phloem and then remobilizes from aged to young leaves, where Zn is loaded into plasmodesmata and stored in the endosperm cavity [29]. Soil Zn fertilization helps in Zn transportation from roots to the above-ground parts of plants and relocalization to grains, which improves plant growth, yield and biofortification of dietary crops (Figure 2).

Alterations in Zn concentration activate different genes to prevent excessive or poor absorption and uptake in plant tissues through transcriptional factors, enzymes, channels, and transporters. Low molecular weight, chelates, and proteins are involved in Zn homeostasis in cytosol, and storage in intracellular compartments. Zinc homeostasis is a complex mechanism in subcellular compartments such as vacuoles (which offers tolerance to Zn toxicity), cell walls, and vesicles. Zn remobilization from the subcellular compartments is crucial during deficiency/senescence or abscission [30,31]. Zinc is an effective nutrient that enhances germination; cell membrane stability; stomatal movement; photosynthesis; and respiration regulation, also interacts with stimulating antioxidant enzymes and protein synthesis, which leads to better plant health and productivity; its scarcity reduces chlorophyll content and protein synthesis [32].

Different strategies are being adapted for the application of Zn fertilizer to plants to come up with an appropriate method, fertilizer form, and dose-response. The most common strategies for Zn fertilization are soil and foliar application, and seed priming, which are viable alternatives that allow plants to survive under Zn-deficient soils by improving crop performance, growth, and yield [33].

### 2.1. Zinc Presence and Functions in Soil

Zinc is the 23rd most abundant element in the earth’s crust, found in polymetallic mines. Zinc in soil can be found in a wide variety of forms including water soluble; organically adsorbed; exchangeable; chelated; and soil solution Zn. Soil parent material is one of the important factors for the bioavailability of Zn in soil. Volcanic rocks contain ~70–130 mg kg^−1^ of soil Zn concentration, while carbonated rocks and sandstone contain ~20 and ~16 mg kg^−1^ soil Zn concentrations, respectively [34]. Soil Zn is available to plants in the form of Zn^2+^, ZnOH^+^, and Zn complexes with soluble organic materials that can be affected by high carbonates, bicarbonates, pH, phosphorous content, and imbalanced macronutrient fertilizers [7,35]. Zinc availability to plants and leaching vary with soil clay and organic matter content. Zinc is not uniformly adsorbed to clay particles, where high Fe and Al oxides, and organic matter contents contribute to the reduction of Zn availability. Soil pH is anti-proportional to the availability of Zn in soil solution for plant uptake. High soil pH renders Zn desorption in clay and organic matter particles, thus decreasing Zn^2+^ phyto-availability [36].

### 2.2. Zinc Fertilization and Sources

Fertilizer management with Zn-containing fertilizers is one of the best options to deal with Zn deficiency in soil and plants. Zinc fertilizers are applied as broadcast and sprayed on soil; banded application; foliar sprays; seed priming; or dipping roots of transplanted crops [33,37]. There are different types of Zn fertilizers; each one is used based on its effectiveness for crops. There is a wide range of Zn fertilizers including Zn sulphate (ZnSO_4_); Zn oxide (ZnO); Zn chloride (ZnCl_2_); Zn-coated urea/superphosphate; and Zn oxy-sulphate. Among these, Zn sulphate and Zn oxides are the most commonly used Zn fertilizers around the world. The use of Zn oxide has recently increased due to its potential advantages to agriculture and especially crop production systems [38,39]. Zinc fertilization has a high response in the soil with low Zn concentration; grain Zn concentration is higher in the soils with low Zn fertility as compared to high Zn fertility soils. 

### 2.3. Zinc Application Methods and Timings

Soil and foliar application of Zn fertilizers are the most studied and commonly used techniques in tropical conditions for enhancing crop nutritional status, growth, and yield. The effectiveness of Zn fertilization is better defined by the time of application. Zinc fertilizer can be applied during planting onto soil and during vegetative, flowering and grain-filling stages via foliar spray [37,38].

Soil Zn application is one of the most used mechanisms for increasing grain Zn concentration and crop productivity. Soil-applied Zn distributes through different transformation mechanisms depending on the soils, crops, and environmental conditions. Agricultural soils reserve numerous Zn fractions depending on leaching from the soil; extractability; plant uptake; adsorption; and association with other minerals. An understanding of the effectiveness of Zn application to soils, plants, and the environment, as well as its distribution and mobility in soil solution, is crucial [40,41]. Soil Zn application is less effective due to low mobility; absorption in soil; high phosphorous fertilizer application; and high carbonate and oxide complexions in tropical agricultural soils, which affect Zn nutrition and productivity [42].

Nutrient foliar application works as a novel solution to address the current challenges of agricultural production systems. Foliar Zn application has received more attention in recent decades due to its benefits of quick absorption by plants, cost-effectiveness, and low influence on soil health dealing with malnutrition [43]. The application of foliar Zn at the vegetative stages of crop plants is absorbed by leaves and quickly used for plant metabolic processes. Its absorption rate can be different for different forms of Zn application to the leaf surface, entering leaves through stomata while transporting via apoplastic and symplastic pathways to different plant parts. Zinc is absorbed by epidermal cells and transported via vascular bundle/phloem to grain tissues. A minute quantity of foliar Zn oxide can improve tolerance against oxidative stress by alleviating and defending the structure of plant cell membranes [44,45].

In addition, the priming of targeted crop seeds in a Zn solution is used for uniform Zn application, which is important for better crop establishment under adverse soil and environmental conditions. It is a sustainable technique that rapidly increases seed establishment and quality attributes that subsequently enhance plant growth. Several studies have reported that seed priming increases seed germination and emergence, and growth by affecting water use efficiency, nutrient uptake and providing resistance to biotic and abiotic stresses [46,47].

The dipping of transplanted crop roots in Zn solution/suspension has a driving impact on the root system, with a stimulative effect on the growth, photosynthetic efficiency, and antioxidant system of plants [48]. All these techniques can contribute to the root absorption and root-to-grain distribution of Zn for better agronomic biofortification. 

## 3. Biofortification

Biofortification is the most sustainable and cost-effective strategy to produce nutrient-enriched crops that sustain human bodies for the long term and eliminate malnutrition. Biofortification is a key tool that generates and releases micronutrient and vitamin-enriched genotypes of different crops by improving their biological functions to supply quality and nutritional food to the general public [49]. Biofortification is a long-run approach with a one-time investment, precise and cost-effective for correcting nutrient deficiency in staple crops. It is considered an ideal intervention that benefits poor and rural communities with nutrient-dense edible grains through agronomic and genetic approaches for sustainable and durable profits [50]. However, our current focus is agronomic biofortification, which is considered the most feasible and sustainable technique to provide a solution to plant-based foods. 

### 3.1. Agronomic Biofortification

Agronomic biofortification is a successful, straightforward, quick, and least labor-intensive technique to provide the general public with nutrient-rich food to overcome micronutrient and vitamin deficiencies [51]. Agronomic biofortification is increasing nutrient accumulation in edible plant tissue through fertilizers or triggering factors. Agronomic biofortification deals with malnutrition through fertilization via soil, foliar application, priming, and seed coating, as well as microbial inoculation, which enables the plants to uptake available micronutrients directly and increase growth and productivity. Agronomic biofortification mediated through mineral fertilizer application via soil or leaves combined with microbial inoculation can improve targeted nutrient availability for plant uptake, increase crop productivity, and nourish plants with more nutrients for better human nutrition (Figure 3).

### 3.2. Mineral Biofortification

Mineral fertilizers are composed of essential minerals that can improve soil characterizations and micronutrient concentration in plants for quality production. The availability of mineral nutrients in the soil can be improved with the application of highly soluble and mobile mineral fertilizers. Mineral biofortification is valuable for a variety of fortified foods including cereals, legumes and vegetables, particularly ingested foods [52,53]. Micronutrient supplementation to staple crops is adopted as a promising strategy to improve micronutrient content in the edible organs of plants all over the world, especially in the tropical regions of developing countries of the world [54]. Cereal crops including rice, wheat, and corn are the most consumed grains in the world and are being adopted as an excellent tool of agronomic biofortification to counter malnutrition [55].

Zinc deficiency impairs the growth and development, biochemical, and metabolic activities of plants, which impacts nutrition and food security [13]. Zinc deficiency in plants normally appears after 2–3 weeks in young and meristem tissues under low-Zn growth conditions. Soil Zn-deficient conditions cause pollen sterility and severely affect shoot-root growth, which leads to low grain yield. Zinc fertilization via soil and leaf are key technique to alleviate Zn deficiency in plants. Zinc fertilization via soil and foliar application improves performance, grain Zn concentration, and the yield of crop plants [37,55]. Soil Zn fertilization allows plants to uptake, translocate, and remobilize Zn in plant stem and grain tissues. Zinc is most commonly applied via mineral fertilizers such as zinc sulfate, zinc oxide, and chelates that significantly enhance plant growth, yield and Zn nutrition. In addition, foliar Zn fertilization counters Zn scarcity in standing crops, and Zn spray at the tasseling and milking growth stages improves grain development and grain Zn concentration [39,56].

Zinc biofortification of cereals reported some excellent results by improving the vigor and nutritional status of plants under limited water and nutrient availability, especially Zn in soil. Biofortification of legumes is an excellent and interesting strategy to improve human nutrition, especially in African countries, where legumes are consumed as a staple food [11]. Zinc biofortification of corn can be managed as a dual-purpose crop, where enrichment of grains with Zn could improve human body Zn concentration, while enrichment of shoots with Zn increases animal feed [57]. Wheat is a staple food of most of the global population, which can better be addressed by Zn biofortification to provide fortified food [58].

### 3.3. Microbe-Mediated Biofortification

Microbial application to enhance Zn availability in soil and plants is comparatively less explored. Biofortification using microbes is an environmentally safe and cost-effective strategy for increasing nutrient bioavailability in dietary food crops by reducing phytic acids in grains [17,22]. In addition, microbes adopt several mechanisms, including acidification; oxidation; reduction; solubilization; and chelation, as well as modify root architecture and physiology, to facilitate nutrient availability and biofortification. Soil microbial activities are responsible for increasing Zn nutritional status by its solubilization and re-localization into plants, thus linking microbes with the higher grain Zn biofortification of cereal crops [8,59,60]. The use of microbes is a possible alternate strategy to convert soil-fixed Zn into available Zn in crop plants to eliminate Zn malnutrition. Several plant growth-promoting bacteria genera of *Azotobacter*; *Azospirillum*; *Bacillus*; *Gluconacetobacter*; *Acinetobacter*; and *Pseudomonas*, and the plant growth-promoting fungi groups of *Arbuscular mycorrhizae*; *Trichoderma;* and *Piriformospora indica,* have been reported to increase micronutrient solubilization in the root rhizosphere of plants in tropical and sub-tropical regions [59,60,61].

The presence of PGPBs in a soil–root ecosystem can reduce the toxic effect of Zn by increasing bioaccumulation in plant tissues through roots, thus improving plant health. This interaction of PGPBs in a soil–root ecosystem adopts several mechanisms, such as phytohormones synthesis; secretion of Zn chelated compounds; prevention of excessive ethylene secretion; nitrogen fixation; and the biogeochemical cycle, to promote nutrient use efficiency through solubilization; bioavailability; remobilization; and translocation of Zn [62,63]. Plant growth-promoting bacteria produce phytohormones like IAA and cytokinin, which influence root biomass and architecture to promote nutrient use efficiency. These PGPBs stimulate multiple direct and indirect mechanisms in the soil rhizosphere to improve plant health, nutrient cycling, and homeostasis. These direct mechanisms include biological nitrogen fixation, nutrient solubilization, and enzyme synthesis, whereas the indirect mechanism includes resistance to pathogen infestation by producing siderophores and antibiotics [58]. Thus, all these mechanisms make PGPBs suitable alternative tools for enhancing micronutrient acquisition from the soil rhizosphere, and crop biofortification of targeted nutrients.

## 4. Zinc Interaction with Root Mechanisms of Plant Growth-Promoting Bacteria (PGPBs)

The plant rhizosphere is an active and confined surrounding of a soil–root interaction that is characterized by microbial diversity and root exudates to facilitate different metabolic processes of plants and microbes [64]. The microbial population of the rhizosphere could benefit the plants through a series of mechanisms including nitrogen fixation; nutrient mobilization; altering and producing phytohormones; increasing stress tolerance; bioremediation; and acting as a bio-control agent [22]. The rhizosphere is colonized by PGPBs, where they share metabolites and an extracellular matrix to establish a biofilm on the root surface and increase the diversity of native microbiota for improving plant health. These rhizosphere microbial populations are used to improve soil health through a direct association with root exudates, microbial biomass, and enzymatic activities in plant root systems, contributing to the alleviation of the negative impacts of nutrient-deficient soils on plant growth [60,65].

Zinc, due to poor native availability in the root depletion zone, is not mobile in the soil and plants mostly take it up by diffusion [66]. Thus, Zn proximity to the roots can either be achieved by exogenous Zn fertilization, or by increasing root growth and surface area. Inoculation of PGPBs (*Arthrobacter* sp. and *Bacillus* sp.) increases the thickness of the root cortex, volume and diameter of vascular bundles, and peri-cycle that improves Zn uptake and biofortification of wheat grains [67]. Inoculation with bacterial strains of *Pseudomonas* and *Enterobacter* sp. was observed in the wheat rhizosphere with a tendency of Zn mobilization and the production of exo-polysaccharides; siderophores; 1-aminocyclopropane-1-carboxylic acid; antimycotic activities; and phosphorus solubilization, thus improving biofortification of wheat grains [68].

### 4.1. Zinc and Inoculation with PGPBs

Plant growth-promoting bacteria can be used as a prospective alternative to less effective synthetic fertilizers, for eliminating Zn malnutrition in a sustainable and environmentally safe manner. Plant growth-promoting bacterial strains like *Bacillus* sp. and *Pseudomonas* sp., generally known as Zn solubilizers, can solubilize unavailable Zn by the production of chelators and secretion of organic acids; amino acids; vitamins; and phytohormones, and oxido-reduction systems and proton extrusion [59,69]. The production of organic acid by microbial strains is one of the main mechanisms of Zn solubilization. Among these organic acids, the production of 2-keto gluconic and gluconic acid by PGPBs is responsible for Zn solubilization [70]. Several genera of PGPBs, including *Pseudomonas*, *Rhizobium*, and *Azospirillum* [22], *Bacillus* sp. [60], *Burkholderia cenocepacia*, and *Arthrobacter* sp. [71], are being recognized and characterized for modulating growth, yield, and zinc biofortification of cereal and legume crops. Some of these PGPBs and their interactions with Zn fertilizer improve grain Zn concentration as summarized in Table 1.

Zinc solubilizing bacteria improved the growth and quality of wheat through siderophore and exopolysaccharide production, Zn solubilization and transportation from soil to above-ground plant. Among these bacteria genera, inoculation with *Azospirillum*, *Pseudomonas* sp., and *Bacillus* increased shoot length; root-shoot biomass; grain Zn bioavailability; and Zn use efficiencies in wheat-maize crops under tropical conditions [8,74]. Therefore, our team developed large field-scale experiments with the integrated use of Zn and PGPBs in the tropical conditions of Brazil. Inoculation with PGPBs and Zn fertilization improved shoot (25–31%) and grain (30–34%) Zn concentration in wheat [83], while the residual effect of Zn fertilization increased shoot (25–30%) and grain (25–26%) Zn concentration in maize, better contributing to agronomic biofortification [78].

### 4.2. Zinc and Co-Inoculation with PGPBs

The global inoculants market has been looking for new strains with new formulations and validations of application methods. In the last decade, the idea of the combined application of different species of microorganisms contributed to different plant processes, known as mixed inoculation or co-inoculation. Currently, a variety of co-inoculants consisting of symbiotic rhizobia are commercially available in the market for many crops [84]. The inoculants of co-inoculation have synergetic interaction with one another that could effectively support plant growth, yield, and nutrient use efficiency by increasing root morphological traits and phytohormone production [85]. Co-inoculation with *Azotobacter* sp. + *Azospirillum* sp. and *Bacillus* sp. + *Pseudomonas* sp. favored growth; the increase in the number of branches; and the productivity of seeds, leaves and essential oil in two cuts of basil plants, contributing to the reduction in the use of mineral fertilizers [86]. Co-inoculation of *Andrographis paniculata* with *Azotobacter chrococcum*, *Bacillus megaterium*, *Pseudomonas monteilii*, and *Glomus intraradices* improved soil chemical properties, phosphatase, plant growth, yield and grain quality [87].

Co-inoculation is an effective strategy dealing with multifaceted plant growth mechanisms in a cost-effective, environmentally friendly, and sustainable manner. Several combinations are so far reported with enhanced growth and quality traits of different crops. Co-inoculation of *Bacillus megaterium* with *Rhizobium* improved the root–shoot biomass of common bean plants and led to the alleviation of biotic and abiotic stresses when compared with single inoculation [88]. Co-inoculation of different *Bacillus* sp. strains enhanced Zn uptake in the wheat–soybean cropping system, leading to higher growth, yield and improved biofortification of wheat and soybean [77]. 

Co-inoculation of *Rhizobium* sp. strains with *Pseudomonas* sp. strains increased the nodulation, chlorophyll content, and nutrient uptake which contributes to higher growth, and quality grain production of different legume crops [89]. Our team reported in a detailed field study that co-inoculation of *Rhizobium tropici* with *Azospirillum brasilense*, *Bacillus subtilis*, and *Pseudomonas fluorescens*, in combination with and without soil Zn fertilization were observed as the most effective combinations for the improved growth, yield, and nutrient use efficiency that elevated Zn uptake in shoot and grains (Figure 4), thus leading to the biofortification and higher estimated Zn intake of common beans to benefit the human, especially Brazilian, population [90].

The consortia or inoculation of *Azospirillum*, *Azotobacter*, *Pseudomonas*, and *Bacillus* sp. inoculants promote plant growth; root-shoot biomass; biochemical and physiological aspects; and enzymatic activities which effectively alleviate stressful conditions, achieve food security and maintain sustainable agriculture practices [91,92]. There exists complex plant–microbial interactions in below-ground regions, such as microbial biodiversity, growth-promoting attributes, mechanisms of action, and interaction with the already existing population, all these are key functions to understand appropriate plant growth and health while maintaining a sustainable agriculture system. Several Zn solubilizing bacteria are being reported as sustainable alternatives that could not only improve Zn content but also enhance phosphorous and potassium solubilization; nitrogen fixation; and production of phytohormones (kinetin, indole-3-acetic acid, and gibberellic acid), as well as the synthesis of siderophores; hydrogen cyanide; and ammonia. All these factors together discourage the use of synthetic fertilizers by promoting plant growth, productivity, and soil fertility status [21,93,94]. Thus, inoculation with PGPBs could be one of the best transmission tools to optimize the harnessing of climate change to better understand the establishment of sustainable agriculture.

## 5. Conclusions and Future Perspective

The study demonstrates the critical interplay between zinc (Zn) mineral nutrition and plant growth-promoting bacteria (PGPBs) in tropical agricultural systems. It underscores the positive relationship between Zn and PGPBs, showcasing how the presence of specific bacteria enhances the absorption and utilization of zinc by plants. This synergistic interaction not only improves plant growth and development but also bolsters the nutritional quality of crops, particularly in Zn-deficient soils, common in tropical regions. It sheds light on innovative approaches to enhancing plant mineral uptake, optimizing nutrient utilization, and fortifying crops with essential minerals, ultimately contributing to improved human nutrition. The positive correlation between PGPB and plants not only facilitates enhanced Zn uptake but also contributes to improved nutrient use efficiency, overall crop performance, and agronomic biofortification. This synergy between Zn and PGPB offers promising avenues for sustainable agriculture, addressing nutritional deficiencies, enhancing crop resilience, and ultimately ensuring food security in tropical regions.

Extensive research has illuminated numerous facets of this field, yielding a wealth of knowledge, yet the continual emergence of new bacterial strains or species exhibiting plant growth-promoting qualities reveals previously unexplored dimensions within ecosystems that could revolutionize agriculture. Despite this understanding, a significant breach persists between laboratory and field experimentations, often acting as a bottleneck hindering the widespread adoption of PGPB in agriculture. Bridging this gap demands not only the involvement of scientists but also a collaboration with business and political stakeholders, to work together to unlock the transformative potential of PGPB, fostering agricultural and economic prosperity while maintaining the commitment to sustainable practices and a healthy environment, especially in tropical regions.

## Figures and Tables

**Figure 1 plants-13-00571-f001:**
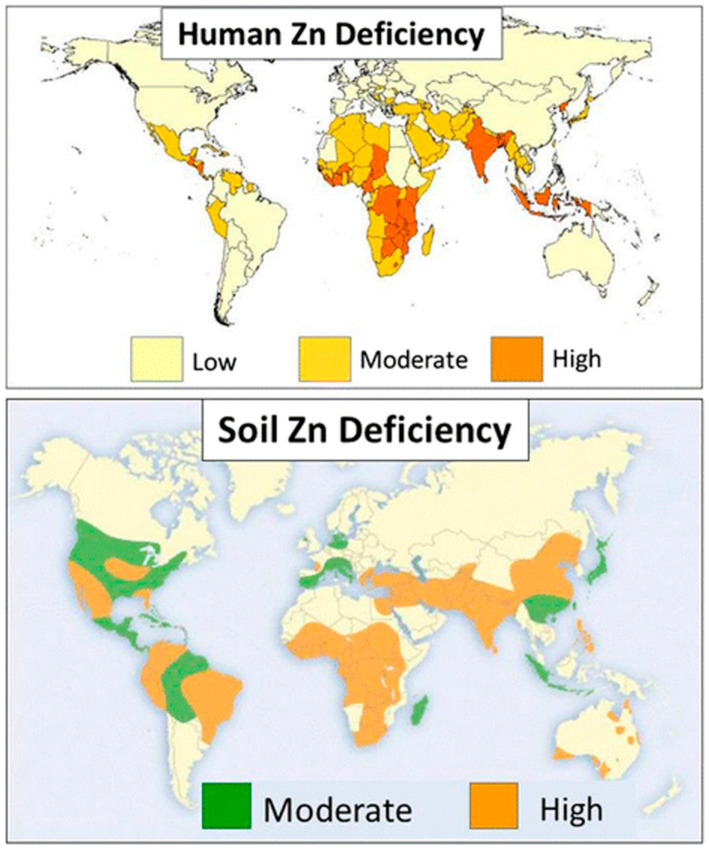
Global distribution of human Zn deficiency and soil Zn deficiency, modified from Wessells and Brown [25], and Alloway [26], respectively.

**Figure 2 plants-13-00571-f002:**
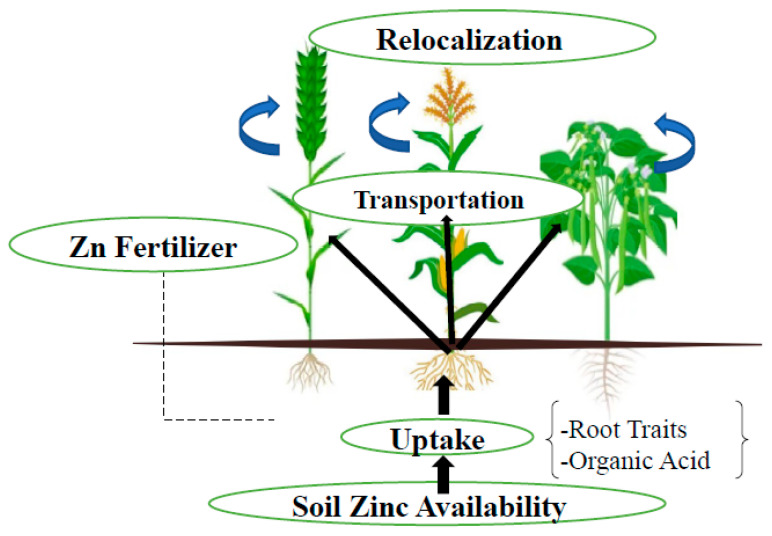
Possible mechanism of Zn transportation and mobilization from soil to grain.

**Figure 3 plants-13-00571-f003:**
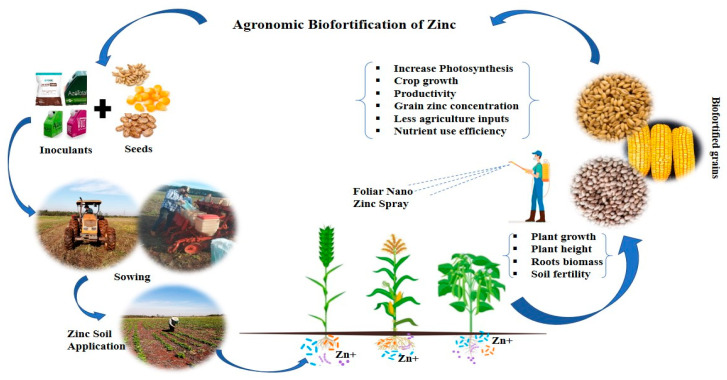
Agronomic biofortification and its outcomes in tropical field experiments. Photos are taken from the original experiment of the first author. The seeds were first inoculated with their respective inoculants, and sown with a drill sowing method; Zn fertilizer and nano Zn oxide were then also applied into the soil. All these approaches together improved crop growth, yield, and Zn concentration in the edible tissues.

**Figure 4 plants-13-00571-f004:**
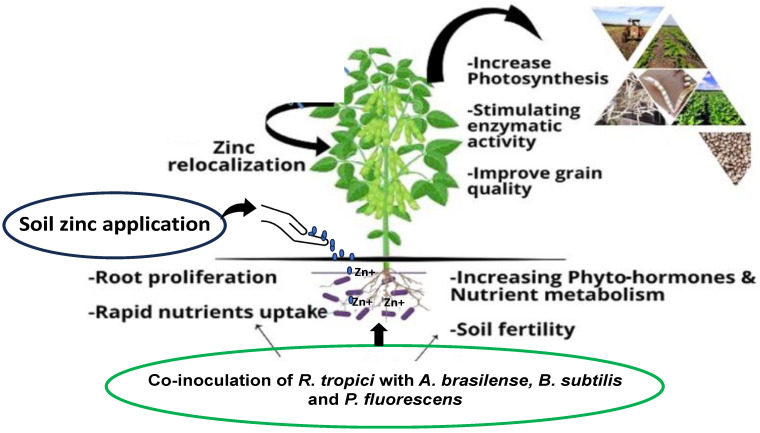
Schematic representation of the most-used plant growth-promoting bacteria and soil zinc application in tropical regions for higher productivity of common beans.

**Table 1 plants-13-00571-t001:** Microbe-mediated biofortification of some important crops.

Crop	Types of Inoculants	Biofortification of Nutrients	References
Wheat	*Pseudomonas* sp.	31% increase in Zn	[72]
	*E. cloacae* subsp. *dissolvens* MDSR9	37% and 21% increase in Zn and Fe	[73]
	*P. fragi* EPS1	Twofold increase in Zn	[74]
	*Exiguobacterium aurantiacum* MS-ZT10	Sixfold increase in Fe and Zn	[75]
	*P. fluorescens* strain Psd	85% increase Zn	[76]
	*Bacillus aryabhattai* MDSR 7	45% increase in Zn	[77]
	*B. subtilis*, *Arthrobacter* sp.	Twofold increase in Zn	[22]
Maize	*P. fluorescens*	Increased Zn uptake by 33–35% in shoot and 37–42% in grain	[78]
	*B. subtilis* ZM63 and *B. aryabhattai* ZM31	68% in Zn	[59]
Rice	*Bacillus* sp. SH10 and *B. cereus* SH17	22–49% increase in Zn translocation to grain	[79]
	*Sphingomonas* sp. SaMR12, *Enterobacter* sp. SaCS20	22% in Zn	[80]
Chickpea & pigeon pea	*P. plecoglossicida*, *B. antiquum*, *E. ludwigii*, *Acinetobacter tandoii*, *P. monteilii*	5–23% in Zn	[81]
Soybean	*E. cloacae* subsp. *dissolvens* MDSR9	33% and 25% increase in Zn and Fe	[73]
	*B. aryabhattai* MDSR 14	36% in increase in Zn	
Common beans	*R. tropici* + *B. subtilis*	Increased Zn partitioning to grain by 11–14%	[82]

## Data Availability

Not Applicable.

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
