# Peer review of "Interaction of Zinc Mineral Nutrition and Plant Growth-Promoting Bacteria in Tropical Agricultural Systems: A Review"

_plants, 2024, doi:10.3390/plants13050571_

Round 1

Reviewer 1 Report

Comments and Suggestions for Authors

The stated objective of Jalal et al.'s review was to investigate, based on the available literature, whether there is a synergistic effect on the growth and productivity of cereal and legume plants of simultaneous treatment with seed inoculation of plant growth-promoting bacteria (PGPBs) and the application of Zinc at both soil and foliar levels.
Honestly, after reading the title and introduction of the manuscript, I expected to find a summary of the most important published results regarding the effects of these two factors (PGPBs inoculation and zinc fertilization) in tropical crop cultivation. However, this focus is only present in Section 7 of the manuscript, and the presented data on the main findings are quite scarce (only one table with a percentage of Zn increase after inoculation with various microbes). My impression is that there are still too few experiments to justify a review of this length on this topic and that results from these experiments are not properly highlighted.

The review is lengthy and, at times, challenging to read because it occasionally delves into somewhat generic topics (plant mineral nutrition, zinc dynamics in soil, etc.), and concepts are often repeated in different sections. Sections 2-6 seem to collectively form an extensive introduction where the main issues of zinc deficiency in tropical agricultural soils are explained, along with its impact on crops and the human population. In sequence, these sections describe: the roles of macronutrients and micronutrient minerals for plant growth; zinc dynamics in soil; zinc deficiency in soils and the human population; mechanisms for zinc utilization by plants; strategies for zinc mobilization in soil; common agricultural practices for zinc fertilization; biofortification techniques adopted to improve plant growth (PGPBs inoculation).
In its current form, I believe the review cannot be accepted for publication and requires a careful major revision in both structure and content.
I suggest the authors consider reducing and possibly merging some of the sections from 2-6 to make the manuscript more readable and to ease the transition to sections 7, which constitute the core of the work. Is it possible to further develop this section and enrich it with data (numbers) from the cited literature?

DETAILED COMMENTS
ABSTRACT
L8-16: very well written, this incipit is very attractive for the reader.
L17-19: “This review outlines the impacts of Zn-PGPB interactions on plant growth, root architecture, and productivity in tropical agricultural systems”, can you provide some numbers in Section 7 (or even in Section 8) about the results obtained on these three issues (plant growth, root architecture and productivity)?
L23: It is concluded…

INTRODUCTION
L36-38: not clear. What do you mean by “those essential micronutrients that can increase the momentum of sustainable agriculture”?
L41-45: I would separate human population deficiency (which can be defined as a silent epidemic)  and soil deficiency. I propose to rephrase the two sentences:
“The above-mentioned deficiency of essential micronutrients especially zinc (Zn) is a silent epidemic, which affects about 17% of the global population and is considered as 5th global health risk 43 factor [6]. On the other hand, around 50% of cereal-cultivable soils show widespread micronutrients deficiencies including Zn deficiency due to high soil pH etc….”.
L46: please rephrase, the subject is “soils” not Zn. Also, the reference should be expressed as a number.

L52-54: not clear. Please rephrase.

L58: …abilities to cope…

L61: Satisfying the requirements…. with high levels of soil fertilizers…

L62: what do you mean by “unprecedented”?

L79-80: not clear. Please rephrase

L86-87: this objective is different from what you reported in the abstract. What is the real objective of your review? To evaluate if PGPBs application can increase Zn in grains of staple food or to evaluate the synergistic effect of PGPBs application and Zn fertilization on plant growth and productivity? Please clarify.

L95: in a review, I would avoid making a hypothesis. You can only try to “investigate (based on the current literature) whether there is a synergistic association of… “

Section 2. Plant mineral nutrition: unlocking plant-soil interactions
Is this section really important for your review?
Can you consider the possibility of reducing this section or even deleting it? Maybe some key concepts of this section could be moved in the Introduction?

Section 3. Zinc and its dynamics; Section 4. Zinc management approaches; Section 5. Zinc management in crop production system via Agronomic approaches.
These three sections are really lengthy and repetitive. I suggest merging them in a single section explaining: the importance of Zn for plant nutrition and Zinc management via fertilizations.
L164-169: this part seems more suitable in the introduction.
L204-209: here you are talking again about Zn soil deficiency. Can you move this part in the introduction?
L213-229: this part deals more with Zn management via fertilization (Section 4)
L233-234: not clear. Please rephrase. What does it mean “by preventing growth and yield under”?
L235-249: is this part really important for your review? I think it should be moved to the beginning of the section on Zn importance for plants

Sections 4.3 and 4.4 should be merged and shortened.

Section 5
In my opinion, this section is out of context in the review. I know that Zn soil content can be managed with agronomic practices but you are investigating the combined effect of PGPBs and Zn fertilizations, so why discuss tillage, irrigation and crop rotations?

Section 6 Biofortification
This section should be more focused on Microbes-mediated biofortification (PGPBs) as mineral biofortification with Zn is a Zn fertilization. My impression is that this section is confusing and repeats some things already written before (human population deficiency, Zn deficiency, mineral fertilizers, etc.).

L394-395: not clear. What does it mean “vitamins enriched in high-quality food”?

Section 7. Zinc interaction with root mechanisms of plant growth-promoting bacteria (PGPBs).
This should be the core of your review and should be expanded as much as possible.
L515: can be used as a…
L530-535: can you provide some numbers about these results?
L540-573: are these studies dealing with the effect of combined Zn applications and inoculation with PGPBs? It seems that they only reported the effect of co-inoculation with PGPBs.

Section 8. Plant growth-promoting bacteria and sustainable agriculture
Also, this section seems a bit out of context in the review. What is the main point here? How PGPBs can improve Zn availability for plants and why this is important for sustainable agriculture practices?

Section 9 Conclusions and Future Prospective ( Perspective)
As in the case of the Abstract, this section is very well written. However, I don’t think that the conclusions outlined here are very well supported by the results shown in Section 7.
I would be more cautious and provide some caveats on the limited number of studies available.

Comments on the Quality of English Language

Overall, the use of the English language is satisfactory (though some sentences may need rephrasing, see detailed comments). However, paragraphs are often composed of short, disjointed sentences, making the discourse less fluid (for example, in sections 2 and 3). If possible, I would ask for the writing to be made smoother with transitional sentences between different concepts.

Author Response

Dear Reviewer,

The authors present sincere gratitude to the reviewer who took time from his busy schedule to help us make this manuscript a better paper. We hope that we have answered every inquiry to your satisfaction and also hope that you will find this version of publishable quality. Hope, this version has met the expectations of the reviewer.

The suggestions and comments provided by the reviewers have immensely helped in improving our manuscript. We appreciate valuable and insightful comments that led to possible improvements in the current version. The authors have carefully considered the comments and tried our best to address every one of them. We hope the manuscript after careful revision has met your high expectations. The authors welcome further constructive comments if any.

The revisions have been approved by all four authors and I have again been chosen as the corresponding author. We provided the point-by-point responses and highlighted them in yellow.

Thanks and kind regards, 

Authors

Reviewer 2 Report

Comments and Suggestions for Authors

MS ID # plants-2818685

Title: Interaction of zinc mineral nutrition and plant growth-promoting bacteria in tropical agricultural system: A review

General comments

I found that the sentences are very frequently too bombastic, especially in the introduction. Furthermore, the title is not reflecting the actual content of the review. Only some part of the introduction (half page) and after the biofortification section (6), section 6.3 from page 11 and onward (four pages) deals with the PGPB issue claimed in the title.

What is the difference between agronomic (chapter 6.3) and mineral (chapter 6.4) biofortification? According the description in the paper none.

Detailed comments

Page 1 line 30: The sentence although bombastic but meaningless. “Agriculture is facing one of the most threatening and challenging food and nutrition security issues.”

Page1, line 31: In addition of what?

Page 1, line 35: … modern bred genetics… ???

Page 1, line 36: “Most of the global population is deficient in those essential micronutrients that can increase the momentum of sustainable agriculture practices for maintaining higher quality and yield of crops.” What is the connection between the two parts of sentence, namely part one: Most of the global population is deficient in those essential micronutrients and part two: that increase momentum of sustainable agriculture practice? Furthermore, what is the momentum of sustainable agriculture practice? Please rewrite.

Page 2, line 45: Authors wrote “Zinc is a widespread micronutrient deficiency…”Zinc is lots of things but Zinc is not a widespread micronutrient deficiency Correct.

Page 2, line 46: By definition Zn deficient soils is deficient in Zn. Zn availability for the plant is different issue. Correct the sentence accordingly.

Page 2, line 50: …while enriching grains of staple crops with Zn,… Uptake and translocation is a tricky issue as a referred paper state. Very frequently, grains are not enriched at all.

Page 2, line 62: Unprecedented meaning that it is never done or known before. Are you sure that is the word you want to use?

Page 2, line 84: What do you mean by the sentence” However, there is a lack of literature on the study of integrated use of PGPBs and soil and foliar Zn in cereals and legumes cropping systems under tropical savannah”. It means no research done or…

Page 2, line 92: In this scenario,… which scenario?

Page 2, line 95: The hypothesis is already proven and written in many published papers. Just search for example Google scholar with the key word: synergetic association of different PGPBs.

Page 3, line 100. “Additionally, the effect of inoculation/co-inoculations and Zn fertilization on Zn use efficiencies for sustainable agriculture in tropical savannah.” The sentence is unfinished.

Page 3, line 113: “Plants adapt several strategies to improve nutrient bioavailability in edible tissues…” Plants are not adapt any strategies to improve nutrient bioavailability in edible tissues.

Page 3, line 145: Plant-soil interaction can regulate around 50% of the total ecosystem carbon dioxide (CO2) as a result of the complex association between root system, litter and soil microbial community that can also optimize the availability of carbon and nutrients in the ecosystem [37].

Wrong: the referred Tian et al article claims “Rhizosphere plant-soil interactions may control as much as 50% of the total CO2 released from terrestrial ecosystems (Hopkins et al. 2013; Schimel 1995).” Underlining from the reviewer.

Big difference…

Page 4, line 156: “Mineral nutrition is a cost-effective strategy…”

Mineral nutrition is naturally occurring inorganic nutrient found in the soil so how can it be a cost-effective strategy? Do you mean mineral nutrient supplementation?

Page 4, line 164: “… cultivatable …” Do you mean cultivable?

Page 5, line 176: The higher phosphorous demand in tropical agriculture is another major constraint that can hinder the availability and functions of several micronutrients, especially Zn, stimulating reactive oxygen species and affecting crop growth and yield [46].

[46] Khatun M.A, Hossain M.M, Bari M.A, Abdullahil K.M, Parvez M.S, Alam M.F. and Kabir A.H 2018. Zinc deficiency tolerance in maize is associated with the up-regulation of Zn transporter genes and antioxidant activities. Plant Biology, 20(4) pp.765-770.

This reference has nothing to do with authors claim.

Page 5, line 181: “… a small amount of Zn is required for the proper physiological, enzymatic and metabolic processes of plants [47].” Zn is a micronutrient so obviously required in small quantity.

Ref 47 and 48 are the same.

Page 5, line 186: “Zinc transportation from soil to plant is a complex factory of different mechanisms with the secretion of phyto-siderophores.” What does this sentence mean?

Page 5, line 195: Figure 2 is surplus. It does not show anything new.

Page 5, line 204: “Zinc deficiency is another factor… “ What do you mean by another factor?

Page 6, line 219: “It was reported that both soil and foliar Zn fertilization had positive impacts on grain Zn concentration (, and grain yield of wheat under irrigated tropical conditions that had ultimately improved Zn intake [56].” Please correct this sentence.

Page 6, line 222: “Some other studies also indicated that soil, foliar and soil + foliar improved grain Zn concentration…” Do you mean soil, foliar and soil + foliar applications?

Page 6, line 223: “…leading to higher grain yield and human Zn nutrition irrespective of the agro-climatic and agronomic inputs [56, 57].” What do you mean here?

Page 6, line 232: “The most common Zn fertilization strategies are soil and foliar application, and seed priming, which are viable alternatives that allow plants to survive by preventing growth and yield under Zn insufficient soils [59].” The sentence is confusing, need to be corrected.

Page 6, line 252: “… or dipping roots of transplanted [65].” Transplanted of what?

Page 7, line 257: “…mostly usually used…” ???

Page 7, line 259: “Zinc fertilization has a high response in the soil with low Zn concentration, grain Zn concentration is higher in the soils with low Zn fertility as compared to high Zn fertility soils.” Explain what do you want to say here.

Page 7, line 263: “Zinc can be extracted from the soil in a solution of different compounds.” This sentence does not belong here.

Page 8, line 322: “However, unawareness and the least accessibility to Zn fertilizers reduced small and poor farmer’s interest.” This sentence does not belong here or has to be modified.

Page 8, line 329: No tillage has disadvantages as well…

Page 11, line 463: …comparatively less explored. Comparatively to what?

Page 11, line 464: “Biofortification through microbes … for increasing nutrient bioavailability in dietary food crops by reducing phytic acids in grains [17, 22].”

PGPBs are for more than reducing phytic acids in grains.

Page 12, line 495: “The microbial population of the rhizosphere benefits…” as not all listed happening all the time it is better to say “could benefit”.

Page 12, line 501: “… are used to recover …” What do you mean by that?

Page 13, line 532: You wrote “Among these bacteria genera …” and after you referred to species.

Page 13, line 545: “…symbiotic rhizobia together with PGPBs…” Symbiotic rhizobia is a plant growth promoting bacteria.

Page 13, line 556: “Several combinations are so far reported with enhancing growth promoting and quality traits of different corps.” So far?

Page 13, line 565: “Co-inoculation with Rhizobium tropici and Azospirillum brasilense and nitrogen doses stimulate root nodulation that contributes to higher growth and productivity of common beans under topical conditions 567 [139].”

How did nitrogen doses interact/prevent/change nodulation frequencies?

Page 14, line 594: “…interactions blow ground regions…” Did you mean interactions below ground regions?

Furthermore, “There exists complex plant–microbial interactions blow ground regions like microbial biodiversity, growth promoting attributes, mechanisms of action”. Microbial biodiversity, growth promoting attributes, mechanisms of action are not complex plant–microbial interactions. Rewrite the sentence.

Page 14, line 604: “The consortia or inoculation of … inoculants…” What is the difference between the consortia and group of inoculants?

Page 15, line 608: “… PGPBs have been introduced against stressed environments…” PGPBs can reduce environmental stress or help plants to overcome stressed environments?

Comments on the Quality of English Language

Many sentences are misleading and need correction (see comments to the authors)

Author Response

(The authors gave the same response as above.)

Round 2

Reviewer 1 Report

Comments and Suggestions for Authors

The authors provided only a generic response to my comments and requests. The revised version of the manuscript should also contain CLEAR tracking changes (deleted parts, modified sentences, new sentences) and not only highlighted parts to indicate where the changes have been made. In this situation it is very time-consuming to check and compare the new version with the old version, and as a consequence, I can't decide on the revision work.
Please, provide a point-by-point response to my previous requests, clearly stating what changes have been made.

Comments on the Quality of English Language

The English language is satisfactory.

Author Response

Dear Reviewer,

The authors present sincere gratitude to the reviewer who took time from his busy schedule to help us make this manuscript a better paper. We hope that we have answered every inquiry to your satisfaction and also hope that you will find this version of publishable quality. Hope, this version has met the expectations of the reviewer.

We provided the point-by-point responses and kept them in italics for your kind consideration.

Thanks and kind regards, 

Authors
